# Brevinin-2 Drug Family—New Applied Peptide Candidates Against Methicillin-Resistant *Staphylococcus aureus* and Their Effects on Lys-7 Expression of Innate Immune Pathway DAF-2/DAF-16 in *Caenorhabditis elegans*

**Hui Xie, Yonghua Zhan, Xueli Chen, Qi Zeng, Dan Chen and Jimin Liang ***

School of Life Science and Technology, Xidian University, Xi'an 710126, China; hxie@xidian.edu.cn (H.X.); yhzhan@xidian.edu.cn (Y.Z.); xlchen@xidian.edu.cn (X.C.); qzeng@xidian.edu.cn (Q.Z.); dchen@xidian.edu.cn (D.C.)

**\*** Correspondence: jimleung@mail.xidian.edu.cn; Tel.: +86-029-8189-1060



**Featured Application: In this study, antimicrobial peptides belonging to the Brevinin-2 family were tested against multidrug resistant bacteria and found to be effective. These substances are expected to be successfully utilized in inflammatory therapy and antibiotic replacement. Furthermore, screening methods for anti-inflammatory drugs carried out using the differential expression of immune regulatory pathways can be widely applied to drug testing.**

**Abstract:** The issue of *Staphylococcus aureus* (MRSA) developing a resistance to drugs such as methicillin has long been the focus for new drug development. In recent years, antimicrobial peptides, such as small molecular peptides with broad-spectrum antibacterial activity and special antibacterial mechanism, have shown a strong medicinal potential. In particular, the Brevinin-2 family has been shown to have a significant inhibitory effect against gram-positive bacteria ($G^+$). In this study, we researched the influence of MRSA on the behavior and survival rate of nematodes. We established an assay of *Caenorhabditis elegans*–MRSA antimicrobial peptides to screen for new potent anti-infective peptides against MRSA. From the Brevinin-2 family, 13 peptides that had shown strong effects on $G^+$ were screened for their ability to prolong the lifespan of infected worms. Real-time Polymerase Chain Reaction (PCR) tests were used to evaluate the effect on the innate immune pathway dauer formation defective (DAF)-2/DAF-16 of *C. elegans*. The assay successfully screened and filtered out four of the 13 peptides that significantly improved the survival rate of MRSA-infected worms. The result of real-time PCR indicated that the mRNA and protein expression levels of *lys-7* were consistently upregulated by being treated with four of the Brevinin-2 family. The Brevinin-2 family peptides, including Brevinin-2, Brevinin-2-OA3, Brevinin-2ISb, and Brevinin-2TSa, also played an active role in the DAF-2/DAF-16 pathway in *C. elegans*. We successfully demonstrated the utility of anti-infective peptides that prolong the survival rate of the MRSA-infected host and discovered the relationship between antibacterial peptides and the innate immune system of *C. elegans*. We demonstrated the antimicrobial effects of Brevinin-2 family peptides, indicating their potential for use as new drug candidates against MRSA infections.

**Keywords:** Brevinin-2 family; *C. elegans*; MRSA; DAF-2/DAF-16 pathway; *lys-7*

## 1. Introduction

Multidrug resistant bacteria, such as *Staphylococcus aureus* (MRSA), severely limits the effectiveness of antibiotics [1,2]. MRSA has the highest mortality rate among multidrug resistant bacteria [3] and affects a variety of hosts, both human and animal [4]. One method of overcoming resistance is the development of new anti-infective agents that activate the host's natural immune defenses against pathogens [5,6]. A series of strategies has been adopted to test new anti-infective agents in whole animal models such as *Caenorhabditis elegans* (*C. elegans*) [7]. Although the common screening method for new drugs using *C. elegans* is now widely recognized by researchers [8], the drug screening still focuses on the level of individual behavior (physiological indicators, mortality, etc.) in nematode infection models. The current understanding is that a continued increase of host-pathogen interactions and bacterial pathogenesis relies on the feedback of the innate immunity system to the external stimuli as a key link in most drug screening studies [9].

According to recent studies, the dauer formation defective-2/ dauer formation defective-16 (DAF-2/DAF-16) pathway of *C. elegans* plays an important role in the resistance of gram negative and positive bacteria [10]. DAF-2 encodes insulin-like receptor homolog and its downstream gene *Age-1/Aap-1* encodes phosphoinositide 3-kinase (PI3K) via the activation of serine/threonine kinases (e.g., phosphoinositide dependent protein kinase-1 (PDK-1), Protein Kinase B (AKT-1, AKT-2)) [11]. Therefore, phosphorylation of the transcription factor DAF-16 prevents its transfer from cytoplasm to the nucleus, thereby inhibiting the transcription of a series of genes [12]. If recombinant insulin-1 (INS-1) combines with DAF-2 or there is a DAF-2 function deficiency, PI3K is not activated [13]. If DAF-16 is not phosphorylated, it will enter the nucleus and regulate the expression of antibacterial genes such as *lys-7*, which is related to immunity and stress [11]. It is obvious that the DAF-2/DAF-16 pathway can be used to study the antimicrobial innate immunity of nematodes [14–17]. Most reports have shown that anti-infective agents increase the survival of *Staphylococcus aureus*-infected worms by inducing the expression of the *lys-7* gene in a DAF-16-dependent manner. These studies highlighted that the *lys-7* antimicrobial gene expression is dependent on DAF-12/DAF-16 mediated signaling [18,19]. Furthermore, it has been shown that c-Jun N-terminal Kinase (JNK-1), kinase germline helicase binding (KGB-1, KGB-2) and extracellular regulated protein kinases (MEK-1) participate in the activation of the innate immunity-linked Forkhead Box (FOXO)transcription factor. Thereby, DAF-16 leads to the increased production of *lys-7* mRNA during a *Shigella flexneri* infection [20].

Antibacterial peptides (AMPs) defend the body against exogenous pathogens. AMPs are an important part of the body's immune system and are generally composed of 10–50 amino acids [21]. It is anticipated that traditional antibiotics such as penicillin will be replaced by new antibacterial peptide drugs [22]. AMPs are expected to solve the problem of drug resistance in pathogenic bacteria [23]. Brevinin-2 is a family of peptides that has been found in the skin of certain Japanese frogs [24]. By 2018, there had been 86 peptides belonging to the Brevinin-2 family added to the Antimicrobial Peptide Database (APD) [25]. This family has 33 AMPs. Amino acid residues of homologous analysis showed that the members of the Brevinin-2 family have significant interspecific and intraspecific structural differences, with the exception of four conservative amino acid residues (Lys15, Cys27, Lys28, and Cys33) [26]. Brevinin-2 peptides derived from the pool frog and belly frog demonstrated inhibitory activity (Minimum Inhibitory Concentration (MIC) < 10 mmol/L) against the gram-negative bacterium *Escherichia coli*, the gram-positive bacteria *S. aureus*, and the fungus *Candida albicans* [27,28]. The hemolytic activity of the Brevinin-2 family is weaker than that of the Brevinin-1 family. Therefore, the Brevinin-2 family has a higher medical research potential [29].

In this study, our aim was to discover the novel anti-infective candidates among members of the Brevinin-2 family. To achieve this, we firstly established a MRSA-*C. elegans* model to identify the effects of MRSA on an individual level in nematode worms. We screened the effective antimicrobial peptides of the Brevinin-2 family that were able to extend the life span of MRSA-infected nematodes [30–34]. Finally, the association between Brevinin-2 family numbers and the DAF-2/DAF-16 pathway of *C. elegans* was verified by the *lys-7* relative expression level (mRNA and protein). We revealed the

important role of the conserved innate immunity signaling pathway of DAF-2/DAF-16 by analyzing host-peptide interactions in the *C. elegans* model [35]. In the future, we hope to discover new broad-spectrum antimicrobial drugs against MRSA.

## 2. Materials and Methods

### 2.1. Bacterial Strains and C. elegans

The methicillin-resistant *S. aureus* strain ATCC33591 (MRSA) and methicillin-susceptible *S. aureus* strain ATCC25923 (MSSA) were grown in Trypticase Soy (Boyao Biotechnology Co., Ltd., Shanghai, China) media at 37 °C, 250 rpm. An *E. coli* strain (OP50) was grown in Luria Bertani (Boyao Biotechnology Co., Ltd., Shanghai, China) broth with streptomycin (Hans Chemical Co., Ltd., Shanghai, China, 100 μg/mL) [36]. The *glp-4/sek-1 C. elegans* strain was propagated on a nematode growth medium (NGM) so as to enhance the sensitivity of nematodes to pathogens and eliminate the interference caused by nematode propagation [37]. The animals in all experiments were age-synchronized and embryos were released by bleaching with alkaline hypochlorite and sodium hydroxide (Hans Chemical Co., Ltd., Shanghai, China). The embryos were then placed on plates with *E. coli* OP50 at 25 °C until all *C. elegans* reached the young adult stage (Stage L4) [38]. It is necessary to note that the animal study procedures were approved and monitored by the School of Life Science and Technology Academic Committee of Xidian University and the Xian Jiaotong University Animal Care and Use Committee.

### 2.2. Preparation of Antibacterial Peptides in the Brevinin-2 Family

A total of 86 peptides from the APD were screened, and 35 were found to be effective against gram positive bacteria (G+). For the following experiments, 13 structurally specific peptides from different subfamilies were screened using multiple sequence alignment. All samples were dissolved in M9 buffer with reference to the minimum inhibitory concentration (MIC) from APD. All of these peptides were synthesized by Hai Mai Science Pte. Ltd (Xi'an, Shaanxi, China, the equipment was ExiProgen from BIONEER). (Figure 1 and Table 1).

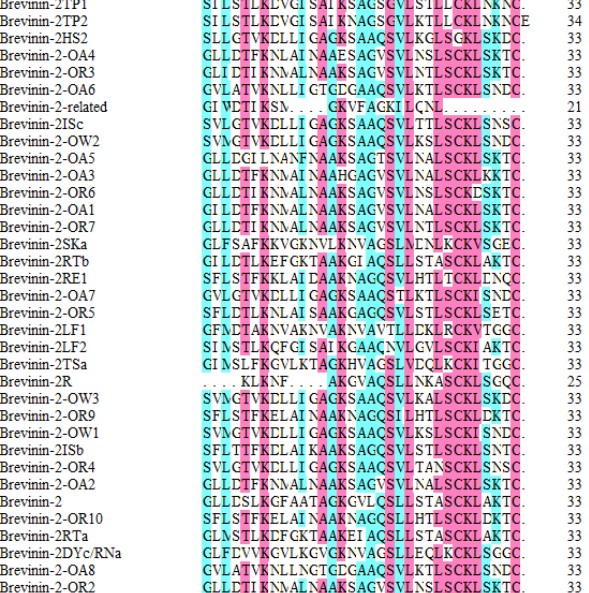

**Figure 1.** Multiple sequence alignment of Brevinin-2 family peptides effective against gram positive bacteria (G+). The sequence marked in red indicates similarity to the corresponding peptide named in the far left column. The sequence marked in blue indicates an even higher similarity to the same peptide. The column on the far right shows the amino acid number of each antimicrobial peptide.

**Table 1.** Gram positive bacteria (G$^+$) effective Brevinin-2 family antimicrobial peptides used in the experiment. MIC: Minimum Inhibitory Concentration; APD: Antimicrobial Peptide Database.

| No. | Name | ADP No. | Peptide Sequences | MIC on APD |
|---|---|---|---|---|
| 1. | Brevinin-2 | AP00075 | GLLDSLKGFAATAGKGVLQSLLSTASCKLAKTC | *S. aureus* (8 µM) |
| 2. | Brevinin-2DYc/RNa | AP00566 | GLFDVVKGVLKGVGKNVAGSLLEQLKCKLSGGC | *S. aureus* (7.5 µM) |
| 3. | Brevinin-2HS2 | AP00881 | SLLGTVKDLLIGAGKSAAQSVLKGLSGKLSKDC | *S. aureus* (19 µM) |
| 4. | Brevinin-2ISb | AP01708 | SFLTTFKDLAIKAAKSAGQSVLSTLSCKLSNTC | *S. aureus* (6.3–25 µM) |
| 5. | Brevinin-2LF2 | AP02476 | SIMSTLKQFGISAIKGAAQNVLGVLSCKIAKTC | *S. aureus* (25 µM) |
| 6. | Brevinin-2-OA3 | AP01836 | GLLDTFKNMAINAAHGAGVSVLNALSCKLKKTC | *S. aureus* (3–13 µM) |
| 7. | Brevinin-2-OR5 | AP01846 | SFLDTLKNLAISAAKGAGQSVLSTLSCKLSETC | *S. aureus* (6.6–13.2 µM) |
| 8. | Brevinin-2-OW2 | AP01853 | SVMGTVKDLLIGAGKSAAQSVLKSLSCKLSNDC | *S. aureus* (3.3–6.6 µM) |
| 9. | Brevinin-2RE1 | AP02612 | SFLSTFKKLAIDAAKNAGQSVLHTLTCKLDNQC | *S. aureus* (20–50 µM) |
| 10. | Brevinin-2-related peptide | AP00599 | GIWDTIKSMGKVFAGKILQNL | *S. aureus* (25 µM) |
| 11. | Brevinin-2SKa | AP01928 | GLFSAFKKVGKNVLKNVAGSLMDNLKCKVSGEC | *S. aureus* (50 µM) |
| 12. | Brevinin-2TP1 | AP02465 | SILSTLKDVGISAIKSAGSGVLSTLLCKLNKNC | *S. aureus* (100 µM) |
| 13. | Brevinin-2TSa | AP00587 | GIMSLFKGVLKTAGKHVAGSLVDQLKCKITGGC | MRSA ($\leq$25 µM) |

### 2.3. Survival Assay of C. elegans

A 24-well plate was filled with MSSA and MRSA in a liquid medium (*S. aureus*/M9 buffer, 1:4, 16 h culture, and 10 µg/mL cholesterol in the culture) and 20 transformed synchronized L4 worms were placed in each well. *E. coli* OP50, *C. elegans*, and M9 buffer served as the control [39]. Each treatment was repeated in at least three replicates. The incubated plate was stored under 25 °C and the survival rate was checked every 12 h. Survival rate was calculated by counting the number of living nematodes remaining in the treatment group at the corresponding time—dead nematodes are dark, stiff, and easy to distinguish from living nematodes.

### 2.4. MIC and MBC Assays Used to Determine Brevinin-2 Family Antibacterial Peptides

The bacteriostatic and bactericidal effects of 13 antibacterial peptide subfamilies were evaluated using the broth micro dilution MIC assay (the method of micro liquid dilution) [7,40,41]. A range of different peptide concentrations (pre-experiment concentration range was based on the G$^+$ effective MIC in the APD) was prepared in a 48-well plate (Table 1). This was followed by inoculation with $10^6$ cfu/mL MRSA and the plate was incubated at 37 °C for 16 h. The number of *S. aureus* colonies was accurately counted using the classical plate coating method. The suspension was diluted to $10^{-3}$, $10^{-4}$, $10^{-5}$, $10^{-6}$, $10^{-7}$, and $10^{-8}$ in 0.1 mL each, and was put into a 37 °C incubator for 72 h. The culture medium was placed on Colony Counter. It was then measured and recorded to determine the exact treatment concentration of *S. aureus*- colony forming units (CFU). If the number of colonies fell outside of 30~300, it was not used. The standard antibiotic Gentamicin for MRSA infections was used as the control. All peptides were assayed at least three times. The recorded endpoint of MIC was the lowest concentration of the peptide used. The peptide showed no turbidity after 16 h of incubation at optical density (OD)600 nm. The minimum bactericidal concentration (MBC) was determined by spreading the culture on sterile agar plates [42]. In this study, the MBC value was the lowest for AMPs of the Brevinin-2 family when no apparent growth of MRSA was observed on the agar plate [43].

### 2.5. Anti-Infective Screening of the Brevinin-2 Family

Antibacterial peptide screenings were performed in a liquid medium in a 24-well plate. The liquid screening medium used was the same as that used in the "*C. elegans* survival rate assay". The culture was supplemented separately with the peptides of the Brevinin-2 family to a final concentration of their MIC for MRSA in this experiment. All peptides of this assay were evenly mixed and 500 µL was transferred to a 24-well plate. In the control wells, the peptide was replaced by 1% dimethyl sulfoxide (DMSO) and *S. aureus* was replaced by *E. coli* OP50. Each peptide was tested three times. Approximately 50 synchronized *glp-4/sek-1* L4 nematodes were transferred to each well and incubated at 25 °C. The survival rate of *C. elegans* was scored manually every 12 h for 7 d. In addition, the turbidity of media was also recorded. A peptide was considered positive if the survival rate of the

experimental group was greater than 60%, while the control group demonstrated a survival rate of less than 20%. All of the positive peptides, based on the results of the first screening, underwent second and third screenings for final confirmation.

## 2.6. Infective Tharepy of Brevinin-2 Family Peptides

"Survival assay of *C. elegans*" was used as reference in this assay. Antibacterial peptide therapy was performed in a liquid medium. The liquid medium used was the same as that in the "*C. elegans* survival rate assay". Approximately 1000 synchronized *glp-4/sek-1* L4 nematodes were transferred and incubated at 25 °C for 12 h with MRSA. The death of infected worms was assessed every 12 h, as detailed above in Section 2.4. Infected nematodes were cleaned three times with M9 buffer. The culture was supplemented with the Brevinin-2 family peptides to a final concentration of their MIC for MRSA. All of the peptides in this assay were evenly mixed and 500 μL was transferred to a 24-well plate. In the control wells, the peptide was replaced by 1% DMSO as a positive control (no therapy). *E. coli* OP50 was used as a negative control. Each peptide was tested three times. Approximately 50 synchronized *glp-4/sek-1* L4 nematodes infected for 12 h were transferred to each well and incubated at 25 °C. The survival rate of *C. elegans* was scored manually every 12 h for 4 d. In addition, the turbidity of media was also recorded. A peptide was considered positive if the survival rate of the experimental group was greater than 60%. The survival rate of the control group was less than 20%. All of the positive peptides, based on the results of the first screening, underwent second and third screenings for final confirmation.

## 2.7. Isolation of RNA in C. elegans

To verify the effect of antimicrobial peptides on the innate immunity system of nematodes, 4 of the positive peptides were selected. Peptide-treated adult worms (100) were collected from the 24-well plate at different times (24 h, 48 h, 72 h, and 96 h) and centrifuged. The same was done with the controls. After collection, the solution with worms was placed in liquid nitrogen for 1 min, followed by the addition of 1 mL of the Trizol reagent. The mixture was then placed in liquid nitrogen 4 times. Next, the supernatant was transferred into an RNase-free tube. The total RNA was extracted in a one-step method by following the instructions provided by Invitrogen. The total RNA was then reverse transcribed into the cDNA using the prime script RT Master kit (Takara, Beijing, China).

## 2.8. Quantitative PCR Analysis of lys-7 in C. elegans

The SYBR Green method and the SYBR Premix Ex Taq II kit (Takara, Beijing, China) were used for real-time RT-PCR. The *C. elegans* gene-specific primers for *lys-7* (40 cycles: 94 °C, 1 min; 60 °C, 1 min; 72 °C, 1 min) and the housekeeping gene *actin* primers (40 cycles: 94 °C, 1 min; 55 °C, 1 min; 72 °C, 1 min) were combined with their PCR mix separately and at a predefined ratio. Amplification was followed by a melting curve analysis. The list of primers used in this study is provided in Table 2. The amplified specific genes were normalized with *C. elegans actin*. Fold changes were calculated using the $2^{-\Delta\Delta ct}$ method. The qPCR analysis was performed with three independent experimental samples in triplicates and gene expression level averages were plotted with the mean $\pm$ SD. The level of significance was analyzed using one-way ANOVA [44].

**Table 2.** Primers for real-time Polymerase Chain Reaction (PCR) used in this study.

| Name | Forward (5′-3′) | Reverse (5′-3′) |
|---|---|---|
| *lys-7* | TTGTTGACTCATCCCTTCC | TGTCCTGCTGGGTTGTAT |
| *actin* | AAGACCACGTCATCAAGG | TTCTCCATATCATCCCAGTT |

### 2.9. Western Blot Analysis of lys-7 in C. elegans

One hundred worms from different treatment groups were frozen separately in liquid nitrogen. They were then homogenized in 2 mL of frozen lysis buffer containing 20 mM of Tris-HCl, pH 6.8, 4 mM glycerol, 1.7 mM SDS, and 30 μM bromphenol blue. The *lys-7* from *C. elegans* were separated on a 15% SDS-polyacrylamide gel. Western blot analysis was performed as previously described [45]. After a sample transfer onto a nitrocellulose membrane, the blot was blocked for 1 h at room temperature with 0.1% Tween-20 and 5% dry milk powder in tris-buffered saline (TBS) (150 mM NaCl, 10 mM KCl, 10 mM Tris-HCl pH 7.6) and then washed with TBS. The blot was incubated with a 1:100 dilution of polyclonal anti-histone *lys-7* antibodies (Haimai Biotechnology Co. LTD., Xi'an, Shaanxi, China). The membrane was left in TBS overnight at 4 °C and washed with 0.1% Tween. "Image Pro Plus" software was used to determine the gray value (IOD). The following formula was used: relative expression of target protein = *lys-7* IOD/*actin* IOD. The results were compared with real-time quantitative PCR. The relative expression levels of genes and proteins were consistent.

### 2.10. Statistical Analysis

Data are shown as the mean ± SD of three independent experiments. All assays were replicated in a comparable manner. Data from the killing assays were analyzed with SPSS 17.0. The pairwise comparison was conducted using the One Way ANOVA significance test. All of the figures were drawn using Graphpad 6.0.

## 3. Results

### 3.1. Assessment of C. elegans Survival Using Liquid-Based Screening

A significant survival rate reduction was observed in *C. elegans* infected with MSSA and MRSA using liquid-based assays (Figure 2) [7]. Worm survival rates dropped sharply between 24 h to 96 h in M9 buffer and *S. aureus* culture. Exposure to MSSA and MRSA took four to five days to kill all *C. elegans*. In a liquid medium, the worms infected by MSSA and MRSA started to die after 12 h of exposure to the pathogen. As MSSA and MRSA induced similar mortality rates in the nematodes, the following experiments were conducted only with MRSA, which is more relevant to clinical and pharmaceutical settings.

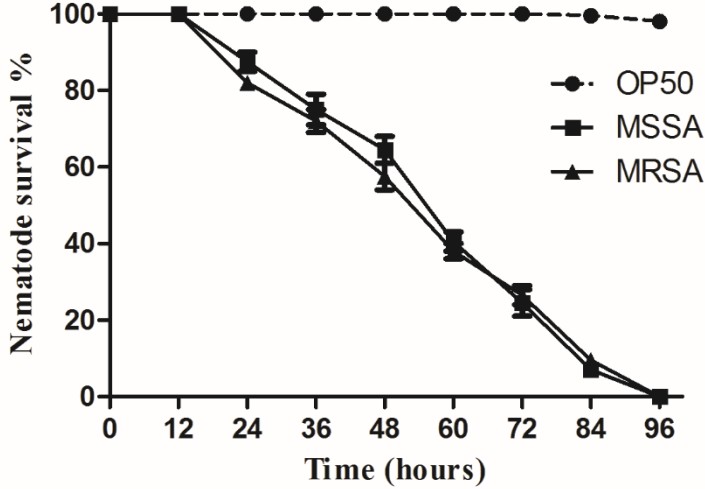

**Figure 2.** The survival of *Caenorhabditis elegans* (*C. elegans*) assessed by liquid-based assay. *Staphylococcus aureus* strain ATCC25923 (MSSA) and *Staphylococcus aureus* strain ATCC33591 (MRSA) were compared to *Escherichia coli* (OP50) as a control group.

### 3.2. The MIC and MBC of 13 Peptides Exhibited Different Anti-MRSA Activity

The MIC and MBC of the antimicrobial peptide tests were used to evaluate the inhibition of MRSA growth. The broth micro dilution test was performed to obtain the MIC and MBC values for 13 peptides. The respective MIC and MBC values of all 13 peptides in the standard medium (TS broth) and the screening medium are presented in Table 3. The DMSO and gentamicin were used as a negative control and a positive control, respectively. The MIC micro dilution test was performed during the screening. Relatively similar values were observed for both media tested and their values, as listed in the APD (Table 3).

**Table 3.** In vitro minimum inhibitory concentration (MIC) and minimum bactericidal concentration (MBC) values effective against *Staphylococcus aureus* strain ATCC33591 (MRSA).

| No | Name | MIC on APD (*S. aureus*) | MIC (MRSA) | MBC (MRSA) |
|---|---|---|---|---|
| 1. | Brevinin-2 | 8 μM | 9.6 ± 1.3 μM | 23.4 ± 4.2 μM |
| 2. | Brevinin-2DYc/RNa | 37.5 μM | 32.3 ± 2.7 μM | 82.5 ± 12.4 μM |
| 3. | Brevinin-2HS2 | 19 μM | 21.5 ± 1.1 μM | 39.7 ± 5.7 μM |
| 4. | Brevinin-2ISb | 6.3–25 μM | 8.7 ± 0.9 μM | 33.8 ± 5.4 μM |
| 5. | Brevinin-2LF2 | 25 μM | 23.4 ± 2.1 μM | 61.5 ± 12.6 μM |
| 6. | Brevinin-2-OA3 | 3–13 μM | 6.7 ± 0.7 μM | 31.5 ± 6.2 μM |
| 7. | Brevinin-2-OR5 | 6.6–13.2 μM | 8.6 ± 1.6 μM | 43.3 ± 8.6 μM |
| 8. | Brevinin-2-OW2 | 3.3–6.6 μM | 4.9 ± 1.2 μM | 28.7 ± 3.9 μM |
| 9. | Brevinin-2RE1 | 20–50 μM | 35.9 ± 4.4 μM | 103.8 ± 14.6 μM |
| 10. | Brevinin-2-related peptide | 25 μM | 27.8 ± 2.6 μM | 58.6 ± 13.4 μM |
| 11. | Brevinin-2SKa | 50 μM | 62.3 ± 6.5 μM | 196.3 ± 26.6 μM |
| 12. | Brevinin-2TP1 | 100 μM | 89.7 ± 11.7 μM | 345.7 ± 64.3 μM |
| 13. | Brevinin-2TSa | ≥25 μM (MRSA) | 31.2 ± 6.1 μM | 78.4 ± 12.5 μM |

### 3.3. Anti-Infective Screening of 13 Brevinin-2 Family Peptides against MRSA

A screening in 24-well plates was performed with MRSA–*C. elegans* to identify potential anti-infective peptides that improved the survival rate of *C. elegans* [46]. Thirteen peptides were screened and demonstrated varying increased survival rates of MRSA-infected nematodes (Figure 3). The adapted standard to classify the positive peptides refers to the "Anti-infective screening of Brevinin-2 family". All of the positive peptides contributed to a *C. elegans* survival rate of more than 60% following infection and treatment for 72 h compared to a survival rate of about 20% in untreated controls (Figure 3). Brevinin-2 and Brevinin-2-OA3 significantly increased the nematode survival rate by nearly 4.2-fold, while Brevinin-2ISb and Brevinin-2TSa increased the survival rate by at least 6.3-fold after 96 h. Only these four peptides increased the nematode survival rate above 60% after 96 h. The control worms were fed on MRSA during the assay.

### 3.4. Brevinin-2 Family Peptides Used for Anti-Infective Therapy

To verify the anti-infective therapeutic potential of Brevinin-2 family peptides, a screening in 24-well plates was performed with MRSA–*C. elegans*. Thirteen peptides were screened and demonstrated increased survival rates of MRSA-infected nematodes (Figure 4). The adapted standard to classify the therapy peptides refers to the "Anti-infective screening of Brevinin-2 family". Four positive peptides contributed to a survival rate of more than 60%. Brevinin-2, Brevinin-2-OA3, Brevinin-2ISb, and Brevinin-2TSa increased the survival rate after 96 h compared to other peptides. The living worms were fed on *E. coli* OP50 during the assay. Four of the antimicrobial peptides identified above affected the nematode survival according to the standard for <60% survival rate. The following experiments were conducted using only these four antimicrobial Brevinin-2 peptides.

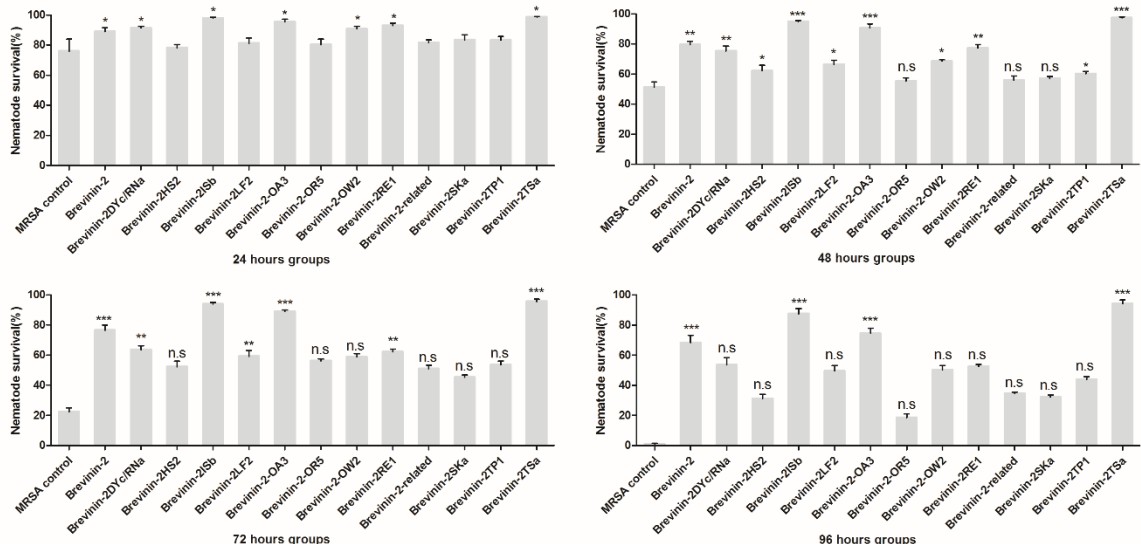

**Figure 3.** Time-dependent anti-infective screening of Brevinin-2 family peptides. $p < 0.05$, 0.01, and 0.001 indicates a significant difference compared with control groups and is labeled with *,**, and ***. If test group rate was below 60%, it was labeled with "n.s".

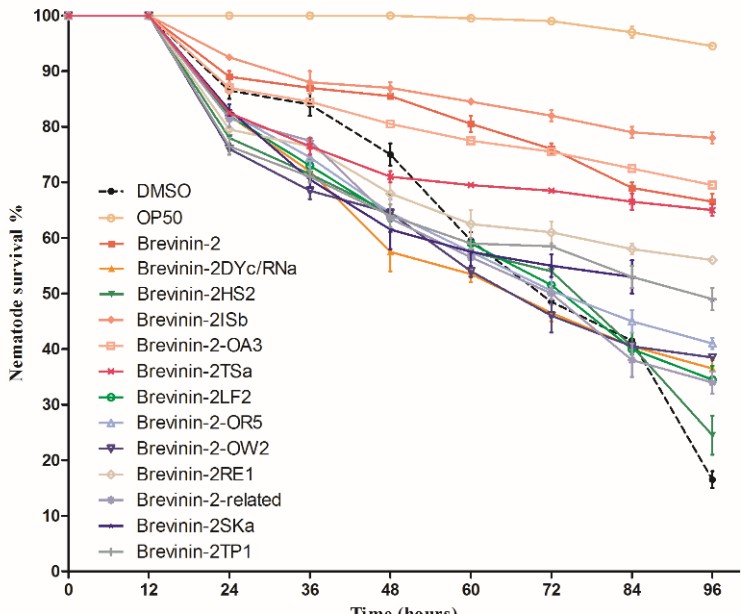

**Figure 4.** The survival of MRSA-infected *C. elegans* assessed by liquid-based assay using Brevinin-2 family peptides.

### 3.5. Confirmation of lys-7 Expression by Real-Time RT-PCR

Assessment of the antibacterial gene expression in *C. elegans*, stimulated by the Brevinin-2 family peptides Brevinin-2, Brevinin-2-OA3, Brevinin-2ISb, and Brevinin-2TSa (the last of which showed the best treatment-related effect), confirmed that these peptides activated *lys-7* expression in innate immunity pathways. The experimental group was treated for 24 h, 48 h, 72 h, and 96 h. OP50 was used as the control treatment. The expression of *lys-7* was consistent. Only Brevinin-2-OA3, Brevinin-2ISb, and Brevinin-2TSa could both inhibit the growth of methicillin-resistant *S. aureus* and play an active role in the innate immunity system (Figure 5). Brevinin-2ISb was the most effective of the group.

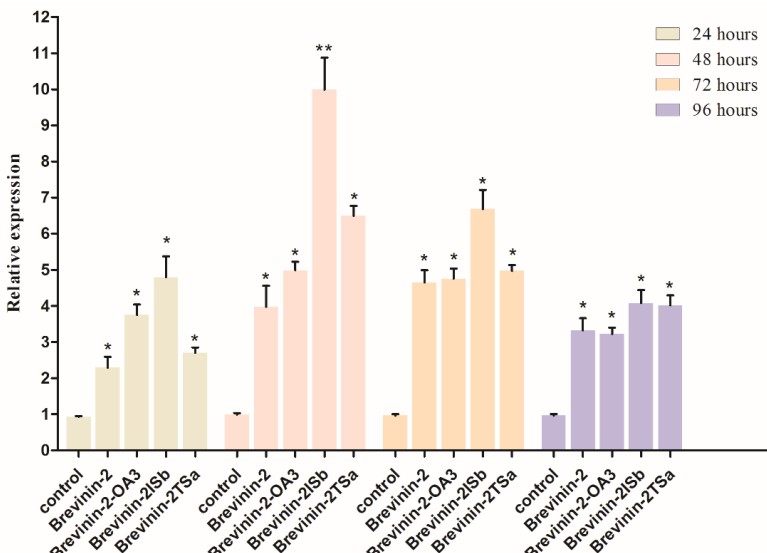

**Figure 5.** The relative gene expression of *lys-7* in *C. elegans* after treatment with four Brevinin-2 family peptides. Compared with control group OP50, * means $p < 0.05$, ** means $p < 0.01$.

### 3.6. Confirmation of lys-7 Expression by Western Blot

The experimental group was treated for 24 h, 48 h, 72 h, and 96 h. OP50 was the control. Western blotting showed differential increase levels of *lys-7* upon exposure to the MIC of Brevinin-2, Brevinin-2-OA3, Brevinin-2ISb, and Brevinin-2TSa (Figure 6). This indicated that the protein expression of *lys-7* was up-regulated in the *C. elegans* that was induced by Brevinin-2, Brevinin-2-OA3, Brevinin-2ISb, and Brevinin-2TSa. The effect of Brevinin-2ISb was the highest (5.987 fold at 48 h).

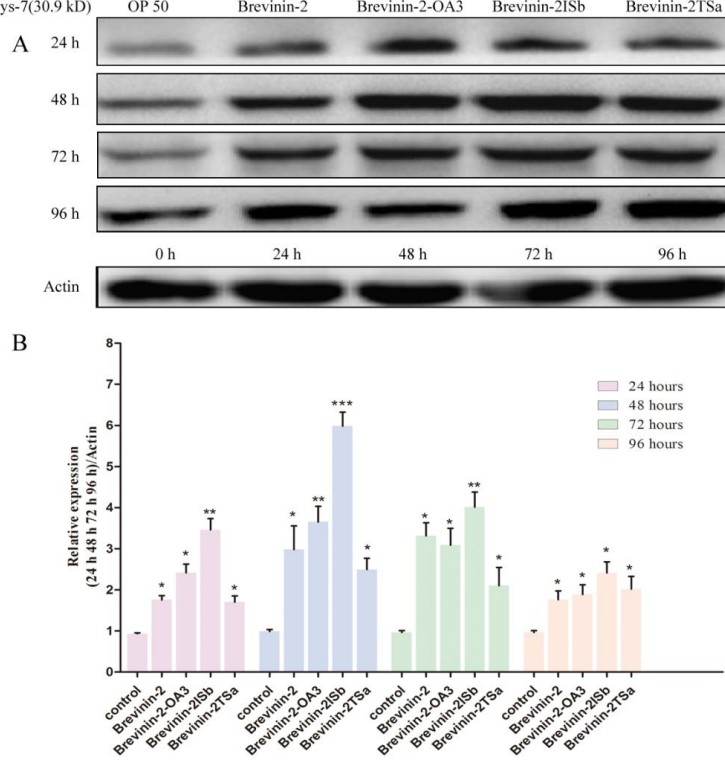

**Figure 6.** The relative protein expression of *lys-7* in *C. elegans* after treatment with four Brevinin-2 family peptides. Compared with control group OP50, * means $p < 0.05$, ** means $p < 0.01$, *** means $p < 0.001$.

## 4. Discussion

In this study, 86 peptides from the APD were tested, and 35 of these were found to be effective against G$^+$. Thirteen structurally specific peptides of different Brevinin-2 subfamilies were screened in the experiments using multiple sequence alignment.

The effects of MSSA and MRSA on nematode movement and mortality had been shown previously. Both were found to have a significant influence on nematode behavior and survival rate. A model of methicillin resistant *S. aureus* infection was designed, and 13 structurally specific peptides from varying Brevinin-2 subfamilies were used for screening. After 96 h of observation, the results showed that all of the 13 antimicrobial peptides had a therapeutic effect on the methicillin-resistant *S. aureus* infected group in comparison with the control group. Brevinin-2, Brevinin-2-OA3, Brevinin-2ISb, and Brevinin-2TSa demonstrated an especially high curative effect on *C. elegans*. Brevinin-2ISb and Brevinin-2TSa were the most effective. Besides the protective effect, all peptides showed bacteriostatic activity. It is possible that nematode survival was—at least in part—improved by the exposure to a reduced number of bacteria or impaired bacteria. Brevinin-2, Brevinin-2-OA3, Brevinin-2ISb, and Brevinin-2TSa also stimulated a markedly increased survival rate after 96 h of treatment compared to other peptides (>60% survival). Our data indicate that these four antimicrobial peptides regulate the innate immunity pathway of nematodes. Brevinin-2ISb has an especially strong therapeutic effect in 96 h, with a survival rate of nearly 80%.

A four-day continuous determination of gene expression was then conducted in order to further study the effect of the peptides on MRSA inhibition and the effect of DAF-2/DAF-16 on the immune system of *C. elegans*. Brevinin-2-OA3, Brevinin-2ISb, and Brevinin-2TSa significantly up-regulated both the mRNA and protein levels of the key antimicrobial gene *lys-7* in the DAF-2/DAF-16 pathway during 24–96 h.

The damage mechanism of antimicrobial peptides to bacterial cell membranes can be roughly divided into two steps. Firstly, antimicrobial peptides selectively adsorb on the surface of a negatively charged bacterial cell membrane, and then sterilize through perforation or non-perforation mode [47]. Some studies have also shown that some antimicrobial peptides can penetrate the bacterial cell membrane into the cytoplasm, affecting the biochemical process of cells, thus playing a bactericidal role in cells [48]. Through the research, we found that the typical antimicrobial peptides of the Brevinin-2 family are more important as immune effector molecules, e.g., host immune defense system regulating, communication between innate immune defense, and acquired immune defense system. The promotion of immune function will eliminate infection and protect the host from pathogenic microorganisms. Previous studies [49,50] have shown that antimicrobial peptides can regulate the production of host cytokines and the antigen-specific immune responses of B cell and T cell, thus initiating innate and acquired immunity. When considering DAF-2/DAF-16, the main innate immune pathway of *C. elegans*, we speculated that the activation mechanism of the four antimicrobial peptides in this study on the innate immune system of *C. elegans* was also induced by regulating the production of host cytokines.

Other studies have shown that some antimicrobial peptides can selectively inhibit the physiological activities of cancer cells without affecting normal cells. For example, the lactoferrin peptide can inhibit the growth of breast cancer cells and gastric cancer cells [51]. Polypeptide drugs not only have a simple structure-activity relationship and can be easily modified, but they also have a similar activity to protein drugs with remarkable curative effects. Therefore, they have great advantages in the process of research, development, production, and use for new drugs. Moreover, many antimicrobial peptide drugs have been listed. For example, daptomycin was approved and marketed as an anionic antimicrobial peptide in 2003. It can be used for skin infections caused by Gram-positive bacteria such as *S. aureus* [50]. Therefore, the four antimicrobial peptides in this study are expected to become new drugs to replace antibiotics after subsequent modification and clinical trials.

Our results imply that the immune system (DAF-2/DAF-16) of *C. elegans* is strongly activated by Brevinin-2, Brevinin-2-OA3, Brevinin-2TSa, and Brevinin-2ISb, the last of which is the most effective. Our data suggest that the survival rate of *C. elegans* depends not only on immune system activation, but also on the MIC of antimicrobial peptides.

Nematode toxicology, screening, molecular biology, and immunology techniques were used to screen and evaluate new peptides as drug candidates. The toxicity of MRSA in nematodes was fully verified, and four infected AMPs were successfully screened for the first time. These AMPs also had a positive effect on the innate immunity system (DAF-2/DAF-16) in *C. elegans.* The mechanism of a peptide-dependent increase in survival rate should be further investigated with consideration of mutant worm strains [34,51].

## 5. Conclusions

In this study, we screened the inhibitory effects of antimicrobial peptides belonging to the Brevinin-2 family on MRSA using APD for the first time. It was observed that four antimicrobial peptides had a therapeutic effect on an MRSA infection in nematodes, these four peptides being Brevinin-2, Brevinin-2-OA3, Brevinin-2ISb, and Brevinin-2TSa, the last of which showed the best treatment effect. It was also found that not only could all four of the aforementioned antimicrobial peptides inhibit MRSA, but they could also activate the innate immunity system related to DAF-2/DAF-16 in nematodes. In summary, this research has laid a foundation for the further study of anti-infective drug screening and innate immunity regulation in *C. elegans* with antimicrobial peptides. The tested peptides in the Brevinin-2 family also present promising prospects for new drug development.

## 6. Patents

Xie Hui, Liang Jimin, Ceng Qi, Chen Dan, Bao Cuiping. A cryopreservation solution and use. Chinese National invention patent. Patent No.: 201610305756.6. (Patent authorization).

Xie Hui, Zhu Shouping, Chen Xueli, Liang Jimin. Based on orthogonal optimization, a rapid and accurate model for infection of *Caenorhabditis elegans* was established. National invention patent. Patent No.: 201710950960.8. (substantive audit stage).

**Author Contributions:** Conceptualization, H.X.; methodology, H.X. and Y.Z.; software, H.X.; validation, Q.Z.; formal analysis, D.C.; investigation, H.X.; resources, J.L.; data curation, H.X.; writing—original draft preparation, H.X.; writing—review and editing, H.X. and J.L.; visualization, H.X.; supervision, H.X.; project administration, J.L., X.C., and H.X.; funding acquisition J.L., X.C., and H.X.

**Funding:** This project was funded by the National Key R&D Program of China (2018YFC0910600), Fundamental Research Funds for the Central Universities JB171207, and supported by the National Natural Science Foundation of China under Grant Nos. 81227901, 61471279, 81530058, and the Natural Science Basic Research Plan in Shaanxi Province of China under Grant No. 2015JZ019, and the Fundamental Research Funds for the Central Universities (NSIZ021402).

**Acknowledgments:** We are grateful to Z. Li, and Y. Sun Research group in Shaanxi Normal University, for kindly providing the technical support and nematodes.

**Conflicts of Interest:** The authors declare that there is no conflict of interests regarding the publication of this paper.

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
