# Peer review of "Brevinin-2 Drug Family—New Applied Peptide Candidates Against Methicillin-Resistant Staphylococcus aureus and Their Effects on Lys-7 Expression of Innate Immune Pathway DAF-2/DAF-16 in Caenorhabditis elegans"

_applsci, doi:10.3390/app8122627_

Round 1
Reviewer 1 Report
Thank you for your responses to my comments. Most have now been satisfactorily addressed in the revised manuscript. There are a few clarifications required, as below:
· Section 2.3: it is still a little difficult to understand exactly how these experiments have been performed. Given nematode survival data in response to exposure to MRSA and MSSA is provided, and this is the method used throughout the rest of the manuscript, I recommend removing this section
· Figure 4: Thank you for the explanation of the lettering system for this figure. Unfortunately it remains difficult to understand. Visually it would be easiest if only those treatments that are statistically significant relative to the MRSA control are indicated with an asterisk, accompanied by an indication of which peptides were most protective in the text (as you have done). With respect to the statistical analysis, how exactly was it performed? Was a correction applied for comparison of multiple groups within one data set? While there are clear differences between MRSA and test groups in the later time points, it is surprising that so many of the treatment groups are significantly different to the MRSA control at 24h post-infection, particularly given the size of the error bars on the MRSA treatment group
· Figure 5: it was indicated that several peptides significantly increased survival in MRSA infected C. elegans. There is no information provided for statistics on this figure.
· Line 284: do you mean “could not only inhibit the…” ?
Author Response
Dear reviewer (professor)
The authors are thankful for the valuable suggestions in the report. The manuscript (Manuscript ID: applsci-393700) has been revised by taking all these suggestions into account, We have corrected them in the manuscript by using the "Track Changes" function in Microsoft Word
1. Section 2.3: it is still a little difficult to understand exactly how these experiments have been performed. Given nematode survival data in response to exposure to MRSA and MSSA is provided, and this is the method used throughout the rest of the manuscript, I recommend removing this section
Answer 1: After careful consideration, we agree with your proposal and removed this section
2. Figure 4: Thank you for the explanation of the lettering system for this figure. Unfortunately it remains difficult to understand. Visually it would be easiest if only those treatments that are statistically significant relative to the MRSA control are indicated with an asterisk, accompanied by an indication of which peptides were most protective in the text (as you have done). With respect to the statistical analysis, how exactly was it performed? Was a correction applied for comparison of multiple groups within one data set? While there are clear differences between MRSA and test groups in the later time points, it is surprising that so many of the treatment groups are significantly different to the MRSA control at 24h post-infection, particularly given the size of the error bars on the MRSA treatment group
Answer 2: Thanks very much for your suggestion and we also thought it is difficult to understand by labeled with different letters. We have followed your advice and changed charts as an asterisk form compared with MRSA control group. (“n.s” in charts means the test group rate below 60%. It’s based on 2.5. Anti-infective screening of the Brevinin-2 family: A peptide was considered positive if the survival rate of the experimental group was greater than 60% )
3. Figure 5: it was indicated that several peptides significantly increased survival in MRSA infected C. elegans. There is no information provided for statistics on this figure.
Answer 3: We are so sorry that our description here is not very rigorous. This experiment was just to prove the anti-infective therapy effect of the four antimicrobial peptides which are more than 60% (2.5. Anti-infective screening of the Brevinin-2 family: A peptide was considered positive if the survival rate of the experimental group was greater than 60%). We had modified this description in the manuscript(line 276-277).
4. Line 284: do you mean “could not only inhibit the…” ?
Answer 4: Yes. We add “only” in line 288. Thanks again. I'm sorry for our carelessness.

Reviewer 2 Report
The paper describes the testing of a group of AMPs, Brevinin-2 family anti-microbial peptides, and a mechanistic elucidation of the promising peptides. The certain peptides showed effective micro molar concentration MIC/MBC, and improved survival rate of C. elegans infected with MRSA as a whole animal model. The results are based on a limited but thorough screening and studies, and indeed interesting, however, it would be informative if the authors describes a possible mode of actions for these peptides to kill the gram-negative bacteria as well as for the increment of Lys-7 expression as an activation of innate immune pathway (DAF-2/DAF-16). Moreover, it would be quite useful if the author discuss how the results could be translated into the applications in other vertebrate organisms and become a scientific lead for new anti-infective agents to replace classical antibiotics.
Author Response
Dear reviewer (professor)
The authors are thankful for the valuable suggestions in the report. The manuscript (Manuscript ID: applsci-393700) has been revised by taking all these suggestions into account, We have corrected them in the manuscript by using the "Track Changes" function in Microsoft Word
1. It would be informative if the authors describes a possible mode of actions for these peptides to kill the gram-negative bacteria as well as for the increment of Lys-7 expression as an activation of innate immune pathway (DAF-2/DAF-16).
Answer 1: Thanks very much for your suggestion and we add related content in “4. Discussion” in paragraph 4 (line 416-430)
Additional discussion is as follows:
The damage mechanism of antimicrobial peptides to bacterial cell membrane can be roughly divided into two steps. Firstly, antimicrobial peptides selectively adsorb on the surface of negative charged bacterial cell membrane, and then sterilize through perforation or non-perforation mode[47]. Some studies have also shown that some antimicrobial peptides can penetrate the bacterial cell membrane into the cytoplasm, affecting the biochemical process of cells, thus playing a bactericidal role in cells[48]. Through the research, we found that the typical antimicrobial peptides of Brevinin-2 family are more important as immune effector molecules:host immune defense system regulating, communication between innate immune defense and acquired immune defense system. Promote immune function to eliminate infection and protect the host from pathogenic microorganisms. Previous studies[49, 50] have shown that antimicrobial peptides can regulate the production of host cytokines and the antigen-specific immune responses of B cell and T cell, thus initiating innate and acquired immunity. DAF-2/DAF-16, the main innate immune pathway of C. elegans, we speculate that the activation mechanism of four antimicrobial peptides in this study on the innate immune system of C. elegans is also induced by regulating the production of host cytokines.
2. Moreover, it would be quite useful if the author discuss how the results could be translated into the applications in other vertebrate organisms and become a scientific lead for new anti-infective agents to replace classical antibiotics.
Answer 2: Thanks very much for your suggestion and we add related content in “4. Discussion” in paragraph 5 (line 431-441)
Additional discussion is as follows:
Other studies have shown that some antimicrobial peptides can selectively inhibit the physiological activities of cancer cells and cancer cells without affecting normal cells. For example, lactoferrin peptide can inhibit the growth of cancer cells, breast cancer cells and gastric cancer cells[51]. Polypeptide drugs not only have simple structure-activity relationship and easy to be modified, but also have a similar activity to protein drugs with remarkable curative effect. Therefore, it has great advantages in the process of research and development, production and use for new drugs. Moreover, many antimicrobial peptide drugs have been listed. For example, daptomycin, as an anionic antimicrobial peptide, was approved and marketed in 2003. It can be used for skin infections caused by Gram-positive bacteria such as Staphylococcus aureus[52]. Therefore, the four antimicrobial peptides in this study are expected to become new drugs to replace antibiotics after subsequent modification and clinical trials.
